# Understanding the Protective Effect of Phytate in Bone Decalcification Related-Diseases

**DOI:** 10.3390/nu13082859

**Published:** 2021-08-20

**Authors:** Pilar Sanchis, Ángel-Arturo López-González, Antonia Costa-Bauzá, Carla Busquets-Cortés, Pere Riutord, Paula Calvo, Felix Grases

**Affiliations:** 1Laboratory of Renal Lithiasis Research, Institute of Health Sciences Research (IUNICS-IdISBa), University of Balearic Islands, 07122 Palma de Mallorca, Spain; pilar.sanchis@uib.es (P.S.); paula.calvo@uib.es (P.C.); 2CIBER Fisiopatología de la Obesidad y Nutrición (CIBERObn), Instituto de Salud Carlos III, 28029 Madrid, Spain; 3Servicio de Prevención de Riesgos Laborales, Gestión Sanitaria de Mallorca, ADEMA University School, 07009 Palma de Mallorca, Spain; angarturo@gmail.com (Á.-A.L.-G.); c.busquets@eua.edu.es (C.B.-C.); pereriutord@gmail.com (P.R.)

**Keywords:** hydroxyapatite, bone resorption, phytate, alendronate, etidronate

## Abstract

Myo-inositol hexaphosphate (phytate; IP6) is a natural compound that is abundant in cereals, legumes, and nuts, and it can bind to crystal surfaces and disturb crystal development, acting as crystallization inhibitor. The adsorption of such inhibitors to crystal faces can also inhibit crystal dissolution. The binding of phytate to metal cofactors suggests that it could be used for treatment of osteoporosis. Our in-vitro study showed that phytate inhibits dissolution of hydroxyapatite (HAP). The effect of phytate was similar to that of alendronate and greater than that of etidronate. This led us to perform a cross-sectional study to investigate the impact of consumption of IP6 on bone mineral density (BMD) in post-menopausal women. Our data indicate that BMD and t-score of lumbar spine increased with increasing phytate consumption, and a phytate consumption higher than 307 mg/day was associated with a normal BMD (t-score > −1). These data suggest that phytate may have a protective effect in bone decalcification by adsorbing on the surfaces of HAP, and a daily consumption of phytate-rich foods (at least one serving/day of legumes or nuts) may help to prevent or minimize bone-loss disorders, such as osteoporosis. However, further studies are needed to gain a better understanding about the mechanism of inhibition of phytate in bone-related diseases (see graphical abstract).

## 1. Introduction

Polyphosphate molecules have been commonly used as a drinking water softener, acting as an inhibitor of calcium salt crystallization since the beginning of the 20th century. Nevertheless, the use of these compounds as the body’s regulators of calcification was not first investigated until the decade of 1960s, when it was first studied by Fleisch et al. Pyrophosphate, a natural polyphosphate, can act as a potent inhibitor of calcification by occupying some of the inorganic phosphate sites on the surface of nascent growing hydroxyapatite crystals, leading to the hypothesis that regulation of pyrophosphate levels could be the mechanism by which bone mineralization is regulated [1,2]. Within humans, pyrophosphate is released as a product of many of the body’s synthetic reactions, and it has been detected in many tissues, including blood and urine [1,2]. However, it was observed that pyrophosphate could inhibit pathological calcification only when it was injected (and not when it was ingested). Several studies demonstrated that pyrophosphate was hydrolyzed and inactivated when orally administered [1,2]. Consequently, bisphosphonates were discovered as chemically stable derivatives of pyrophosphate. Bisphosphonates are a group of synthetic polyphosphates that were shown to prevent pathological calcification even when they are administered orally to animals [3].

Crystallization inhibitors are a group of molecules that bind to crystal surface, avoiding or disturbing crystal growth. The adsorption of such compounds to crystal faces can also inhibit crystal dissolution. Consequently, bisphosphonates could inhibit hydroxyapatite crystal dissolution [4,5] and bone resorption [4,5,6] when they are orally administered. Like their natural analogue pyrophosphate, bisphosphonates have a very high affinity for bone mineral because they bind to hydroxyapatite crystals. Therefore, the retention of bisphosphonates on bone surface depends on the availability of hydroxyapatite binding sites. To date, many studies with in-vitro systems, animal models, and clinical trials have shown that a variety of bisphosphonates can inhibit bone resorption [6,7,8].

Phytate (myo-inositol hexaphosphate, IP6) is a naturally occurring compound that is ingested in significant quantities by humans with diets rich in whole grains, including legumes, whole cereals, and nuts. Consequently, phytate is readily accessible to people consuming a balanced diet (1–2 g/day). Phytate is has been found in all mammalian organs, tissues, and fluids [9,10], and its levels depend on the exogenous supply either orally [9,10] or topically [11,12]. In this sense, when phytate is absent from the diet, the urinary concentration of phytate decreases to undetectable levels after 22 days [9,10]. Additionally, several studies have indicated that phytate is a powerful inhibitor of calcium salt crystallization in urine and soft tissues [13,14], like other polyphosphates, such as pyrophosphate and bisphosphonates [15]. Additionally, several studies indicate that phytate prevents the formation of pathological calcifications in vivo, such as renal calculi [16,17], dental calculi [18], and cardiovascular calcification [19,20,21].

Bone decalcification, which happens during osteoporosis, may be considered the opposite of calcification. As it has been indicated previously, some agents that prevent pathological calcification, such as pyrophosphate bisphosphonates, also inhibit bone decalcification. This effect could be explained, almost in part, because they have a high affinity to bind onto the calcium of hydroxyapatite crystals by chemisorption, hindering both crystallization and redissolution. Some authors have suggested that phytate exhibits effects like those of bisphosphonates on bone resorption [22,23,24,25,26]. In this sense, a study examined the influence of consumption of the calcium-magnesium salt of phytate on the characteristics of bones in ovariectomized rats (an animal model for postmenopausal osteoporosis) [22]. In this study, bone mineral density (BMD) was significantly greater in femoral bones and L4 vertebra in rats fed with phytate-rich diet compared to the rats fed with a diet without phytate after 12 weeks [22]. Consequently, phytin (calcium magnesium salt) consumption has demonstrated to reduce bone mineral-density loss due to estrogen deficiency, and phytate could exhibit effects like those of bisphosphonates on bone resorption [22]. In small cohort studies, phytate has been correlated with bone mass in post-menopausal women [23,24]. In cell-tissues studies, phytate has demonstrated to inhibit osteoclastogenesis [25,26]. Nevertheless, more studies are needed to confirm these findings and to elucidate the mechanism of inhibition of phytate in decalcification process.

Bone decalcification or bone mass loss is mainly an age-related problem, with a higher prevalence among postmenopausal women [27], but bone decalcification is also influenced, among other factors, by nutrition [28]. Diet is a modifiable factor in preservation of bone mass. Several studies have revealed that the low incidence of osteoporosis in some Mediterranean countries could be due to their healthy dietary patterns [29,30,31]. The Mediterranean diet is characterized by a high intake of vegetables, fruits, cereals, nuts, legumes, and olive oil. The beneficial effects of adherence to a Mediterranean diet and some of its components (such as fruits, vegetables, and fish) on bone mass density have been previously studied [29,30,31,32]. However, the effect of the consumption of phytate-rich foods (such as legumes and nuts) on bone mineral density have not been deeply investigated.

Consequently, the remarkable ability of phytate to bind calcium faces of crystals and the previous studies that suggest the possible protective role of phytate on bone loss motivated us to study whether phytate could had a protective effect on bone decalcification process both in vitro and in vivo. Thus, the specific aims of this paper are: (i) to examine the effects of phytate and two bisphosphonates (etidronate and alendronate) on in-vitro dissolution of hydroxyapatite; and (ii) to investigate the association between phytate consumption and bone mineral density of lumbar spine (BMD) in a cross-sectional study with Spanish postmenopausal women.

## 2. Materials and Methods

### 2.1. In-Vitro Study: Effects of IP6 on Dissolution of HAP

#### 2.1.1. Adsorption of Phytate and Bisphosphonate on HAP Crystal

Hydroxyapatite (HAP) suspension were prepared by mixing 130 mg of HAP in Tris buffer (400 mL 0.05M) at pH 7.4 [33]. Immediately, nothing, phytate (as sodium salt), etidronate (in acid form), or alendronate (as sodium salt) was added to HAP suspension and incubated during 8 h and 37 °C with stirring in a thermostatic bath to ensure the drug absorption on HAP crystal surfaces. After 8 h, the suspensions were filtered through a 0.45-micrometer pore filter, and the HAP-adsorbed drug was dried in a desiccator at 25 °C. All crystallization inhibitors were studied in a final concentration of 1 and 3 µM.

#### 2.1.2. Inhibition of HAP Dissolution

The effect of phytate and the two bisphosphonates on the inhibition of hydroxyapatite dissolution was studied as it has been previously described [33]. Firstly, the HAP-adsorbed drug was mixed and resuspended in acidic condition to induce the dissolution of HAP (200 mL 0.15 M acetate buffer) at pH 5.0. The suspension was stirred during 24 h at 37 °C. At different times, aliquots were removed, filtered, and analyzed for dissolved calcium by inductively coupled plasma atomic emission spectroscopy (ICP-AES, Perkin-Elmer Optima 5300DV). All experiments were done by triplicate. All reagents were purchased from Sigma-Aldrich or Fluka.

### 2.2. Cross-Sectional Study: Association of Phytate and BMD

#### 2.2.1. Subjects and Methods

The study was prospective in nature. Four-hundred and forty women were recruited prospectively and consecutively from among those who attended the bone health promotion program carried out within the global health promotion program of the Prevention Health Service (Palma de Mallorca, Spain). Briefly, patients were included if they reported being menopausal at least 2 years ago by self-report (at least 24 months without menstruation) and they agreed to participate in the study and provided the informed consent. Women were excluded if they were in a treatment with bisphosphonates or other drugs used to treat osteoporosis. Ten women refused to participate in the study, and fifteen did not complete the study and were excluded for the lack the data.

Women who satisfied the criteria and signed the informed consent were evaluated with in-dept clinical interviews, including whether they had entered menopause, and hormone concentrations were measured in the case of doubt. Personal and clinical data were collected by GESMA’s Occupational Health Service.

The study protocol was approved by the Balearic Research Ethics Committee of the Balearic Islands (n° IB 4383/20). Written informed consent was obtained from all study participants.

#### 2.2.2. Dietary Assessment, Estimation of Phytate Intake, and Mediterranean Diet Score

A validated 14-item questionnaire for the estimation of adherence to the Mediterranean Diet, PREDIMED (PREvención con DIeta MEDiterránea) [34], was registered for all the included women during an interview with a dietitian. Briefly, for each item was assigned score 1 and 0; PREDIMED score was calculated as follows: score 0–5, low adherence; score 6–9, average adherence; score > 9, high adherence [34]. Estimated phytate consumption was estimated as previously described [35], based on the intake of legumes and nuts, the serving size of each item [36], and the phytate concentration of each item based on published sources [37,38,39,40].

#### 2.2.3. Determination of Bone Mineral Density

BMD of the L1-L4 column was determined to all women by DXA (Ge Lunar Prodigy Advance Bone X-ray Densitometer) and is reported as g/cm^2^. All densitometry measurements were performed by a single technician to avoid interobserver bias. Based on test-retest using 10 subjects, the coefficient of variation for BMD values were 1.7%.

### 2.3. Statistical Analysis

Data are presented as means and standard deviations, means and standard errors, or numbers and percentages, unless otherwise stated. Patients were divided in three groups according to tertiles of phytate intake: low (L: 0–199 mg/day), moderate (M: 200–325 mg/day) and high (H > 325 mg/day). Normality plots (histogram and Q-Q plot) and Kolmogorov–Smirnov test were used to study if data were normally distributed. The homoscedasticity between groups were studied with Levene’s test. Intergroup comparisons of phytate groups employed one-way ANOVA and Bonferroni as post-hoc test for quantitative data; and the chi-square test for categorical variables. According to BMD, patients were divided in two groups according to the t-scores values: low-BMD (t-score ≤ −1) and normal-BMD (t-score > −1). Intergroup comparisons of BMD groups employed the independent-samples *t*-test and the chi-square test or Fisher’s exact test for categorical variables. ROC curves of quantitative risk factors associated to low-BMD were performed. The optimal cutoff values were determined by the maximum Youden index (J), defined as sensitivity + specificity − 1. Univariate and multivariate binary logistic regression models were fitted to assess the association of phytate intake and phytate-rich foods with low-BMD (t-score ≤ −1) and with high BMD (t-score > −1) as the reference (odds ratio (OR) = 1). A two-tailed *p*-value less than 0.05 was considered statistically significant. Statistical analyses were performed using SPSS 25.0 (SPSS Inc., Chicago, IL, USA).

## 3. Results

### 3.1. In-Vitro Study: Effects of IP6 on Dissolution of HAP

In our in-vitro study, HAP was firstly incubated with either nothing (control), phytate, alendronate, or etidronate to ensure that the molecules of polyphosphates bound to HAP crystal surfaces. After that, the incubated HAP was exposed to acid-driven dissolution, and released calcium levels were determined. Figure 1 shows the kinetic curve over the 24-h dissolution period for these three compounds and control. As can be seen, HAP incubated with phytate, alendronate, or etidronate showed less dissolution than HAP incubated with nothing (control).

The inhibitory effects on HAP dissolution of phytate, etidronate, and alendronate are represented graphically in Figure 2. As can be seen, the inhibitory effect of phytate on HAP dissolution (1 µM: 13.2 ± 0.6%; 3 µM: 26.7 ± 0.4%) was greater than that exhibited by etidronate (1 µM: 10.3 ± 0.7%; 3 µM: 14.6 ± 0.7%) and similar to that by 1 µM alendronate (1 µM: 11.4 ± 2.6%; 3 µM: 23.2 ± 1.4%) for both studied concentrations. Furthermore, phytate inhibited HAP dissolution in a concentration-dependent manner, with the percentage of inhibition at 13.2 ± 0.6% at 1 µM and 26.7 ± 0.4% at 3 µM (*p* < 0.05).

### 3.2. Cross-Sectional Study: Association of Phytate and BMD

A total of 415 women completed the study, with a mean age and BMI of 55.7 ± 2.8 and 25.0 ± 4.3 kg/cm^2^, respectively. The main characteristics, bone parameters, and dietetic factors among tertiles of phytate intake are shown in Table 1. As can be seen, women in the highest tertile were significantly younger and had a lower BMI, hip circumference, and body fat mass in comparison to the other tertiles. The adherence to Mediterranean diet increased with the tertiles’ phytate intake (L: 8.2 ± 1.6; M: 8.9 ± 1.5; H: 9.7 ± 1.9; *p* < 0.001). Regarding bone parameters of lumbar spine, bone mineral density and t-scores significantly increased with phytate intake. Furthermore, a high percentage of participants with low-BMD status was found in the first phytate tertile compared to the other tertiles (L: 91.0%; M: 74.8%; and H: 20.4%; *p* < 0.001).

Dietary characteristics between participants with low-BMD and normal-BMD were explored (Table 2). Women with low BMD were older (56.1 ± 3.1 versus 55.0 ± 2.3 years; *p* < 0.001) and had a higher BMI, hip circumference, waist circumference, body fat mass, and blood pressure compared to women with normal BMD. Regarding dietary parameters, women with low BMD had a lower adherence to Mediterranean diet (8.6 ± 1.7 versus 9.5 ± 1.8 points; *p* < 0.001), with a lower consumption of vegetables, fruits, legumes, nuts, and butter than those with normal BMD. No differences were found in fish, meat, olive oil, wine, carbonated drinks, and pastries and sweets. Estimated phytate intake was also lower in low-BMD group than in normal-BMD group (202 ± 110 versus 447 ± 174 mg/L; *p* < 0.001; Figure 3A).

ROC curves and Youden index were calculated for estimate the optimal cut-off value of phytate intake associated with low BMD (Figure 3B). As can be seen, an estimated phytate intake lower than 307 mg/day was the optimal cut-off value, associated to a low BMD with sensitivity and specificity of 81.9% and 80.7%, respectively.

The odds ratio and 95% CI (crude and after adjusting for potential cofounders) for the association of estimated phytate intake (tertiles) and phytate-rich foods (servings of legumes and nuts) with low BMD (versus normal BMD as reference) are shown in Table 3. We observed that a higher phytate consumption was inversely associated to low BMD when comparing the O.R. (95%CI) of second and third tertiles (M: 0.32 (0.16–0.66) and H: 0.03 (0.01–0.06)) using the first tertile as reference (OR = 1) after adjusting for potential confounders. Similar results were obtained for the consumption as servings/week of legumes and nuts: a higher consumption of legumes (O.R: 0.44 (0.33–0.59)) and nuts (OR: 0.39 (0.32–0.47)) were inversely associated to low BMD (*p* < 0.001).

## 4. Discussion

The present study is the first to demonstrate that pre-incubation of HAP with phytate was able to inhibit acid-driven HAP dissolution in a concentration-dependent manner and, therefore, on the bone decalcification process. The effect of phytate and alendronate on inhibiting HAP dissolution was similar and higher than etidronate. Furthermore, in the current cross-sectional analysis, we reported a significant association between low phytate consumption (<307 mg/day) and low BMD at lumbar spine. These results suggest that a diet rich in phytate could protect against bone loss mechanisms.

Our in-vitro results indicate that phytate can inhibit or disturb the decalcification process by adsorbing on the surfaces of HAP crystal and the consequent inhibition of HAP dissolution (such as the bisphosphonates). Therefore, we suggest that phytate may play a role in treatment and prevention of bone-related diseases. Phytate is a natural polyphosphate compound with a high affinity for divalent cations, such as calcium. In this paper, we demonstrate that phytate can act on bone by the physicochemical inhibition of crystal dissolution with an action like alendronate and higher than etidronate. Recent studies have indicated that phytate inhibits osteoclastogenesis on RAW 264.7 monocyte/macrophage mouse cell line and on human primary osteoclasts cell line [25,26]. Other authors also demonstrated that ingestion of phytate also generates other inositol phosphates (IP5, IP4, IP3, IP2), which can also play an important role on bone resorption [16,41]. Thus, phytate may represent a novel type of therapeutic agent of decalcification process by inhibiting osteoclasts activity and HAP dissolution. These facts prove that phytate can be useful for the treatment of bone-related disease. These data are in accordance with the recent studies that indicate that patients with osteoporosis consumed lower amounts of phytate-rich products compared to a population without osteoporosis [24]. Furthermore, studies with animals have observed that when phytate was orally administered, the levels in bone achieved consistent levels, but this concentration decreased drastically to undetectable levels when phytate was eliminated from the diet within 15–22 days [9,10].

In the cross-sectional study, the BMD and t-score values of lumbar spine increased with increasing legumes and nuts intake and, consequently, phytate consumption. Similar results have been observed by other authors. In a cross-sectional study with 433 women, body weight and low phytate consumption were the risk factors with greatest influence on BMD of lumbar spine and femoral neck [42]. In another study with 157 postmenopausal women, the 10-year fracture probability was also significantly higher in the low-phytate group compared to the high-phytate group, both in hip and major osteoporotic fracture [23]. However, the mechanisms underlying the association between BMD and phytate consumption are not completely understood. Osteoporosis is a multifactorial, age-related problem that can be influenced by various food and nutrients intake [28]. It is well known that the Mediterranean diet is associated with a higher BMD [29,30,31,32] and a lower risk of fracture [43,44]. These results could be explained, almost in part, by the phytate-rich food components of this diet: legumes and nuts. Interestingly, the Mediterranean diet results in an intake of approximately 1 g of phytate per day [35], and our study indicates that a consumption higher than 307 mg/day is associated with a normal bone mineral density. Considering that a serving of legumes or nuts contain between 300 and 900 of phytate [34,35,36,37,38,39,40], one serving/day of some of these foods could be enough to ensure this phytate consumption [35,36,37,38,39,40]. Nevertheless, large, long-term, prospective, clinical studies must be performed to assess the benefits and risks of phytate consumption more completely in bone-related diseases.

Osteoporosis is an important worldwide bone disease characterized by low BMD with a consequent increase in bone fragility and susceptibility to fracture. Bisphosphonates are commonly used in the treatment of osteoporosis to reduce fracture risk and improve BMD. Bisphosphonates are compounds with a general P-C-P structure. This structure permits a great number of possible modifications, especially by modifying the two lateral chains in the carbon atom [45,46,47]. Each bisphosphonate has its own physicochemical and biological characteristics. They decrease bone loss, increase BMD, and decrease bone turnover [7,45,46,47]. Consequently, they are administered in diseases with elevated bone destruction, such as Paget’s disease, metastatic bone disease, and osteoporosis. In the latter, they diminish both vertebral and non-vertebral fractures [46]. The bisphosphonates have shown various mechanisms of action in bone that can be classified in two main groups: the physicochemical effects on bone salt crystals and biological effects on bone mineralization and bone resorption. Regarding physicochemical effects, bisphosphonates inhibit the formation and aggregation and slow down the dissolution of calcium phosphate crystals. These effects are related to the marked affinity of these compounds for solid-phase calcium phosphate to which they bind strongly on the surface [7,45,46,47]. Regarding the biological effects on bone mineralization and bone resorption, four mechanisms on the osteoclast activity appear to be involved: inhibition of osteoclast recruitment, inhibition of osteoclastic adhesion, shortening of the life span of osteoclasts due to earlier apoptosis, and inhibition of osteoclast activity. It was found that nitrogen containing bisphosphonates can inhibit farnesyl pyrophosphate synthase, which leads to apoptosis in osteoclasts, the bone-resorbing cells. In contrast, some non-nitrogen-containing bisphosphonates, such as etidronate, can be incorporated into the phosphate chain of ATP-containing compounds, provoking apoptosis and inhibition of bone resorption [7,45,46,47]. Regarding BMD, alendronate, ibandronate, risedronate, and zoledronic acid have been shown to increase BMD by 5–7% and 1.6–5% in the spine and femoral neck, respectively, after three years of treatment [48].

Respecting adverse events, oral administration of bisphosphonates, especially those containing a nitrogen atom, can be accompanied by disturbances in the upper gastrointestinal tract [45,47,49]. Some authors have indicated that phytate can negatively affect the bioavailability of calcium, iron, and zinc. In this sense, a diet with high levels of phytate (as sodium salt) caused rickets in dogs by reducing calcium absorption [50]. Since that time, several studies have labeled phytate as an “antinutrient” to phytate because it passes through the gut and binds to divalent cations (iron, zinc, and calcium), and they are not well absorbed in the intestine. [50,51,52,53,54]. However, several studies indicate that the intake of phytate (as calcium-magnesium salt, phytin) in amounts that represent approximately 0.1% of a total balanced diet has no adverse effect on mineral bioavailability [55,56]. Studies using humans reported that a phytate intake of 2 g per day did not affect mineral balance [57,58,59], and as our results indicated, a consumption higher than 307 mg/day of phytate could protect against bone loss. Consequently, the use of phytate on bone-related diseases can be an alternative to bisphosphonates in those patients who exhibit digestive tract disturbances.

Certain limitations of our study should be mentioned. First, the lack of BMD determination in other body sites, such as total femur and femoral neck, did not allow us to know if this association between phytate and BMD in lumbar spine is extrapolated for other body sites. Second, the absence of data about estrogen levels and lifestyle parameters that can act as potential confounders in the observed association of BMD with phytate is another important limitation. Third, this study was carried out in Mediterranean postmenopausal women, and our results cannot be generalized to other populations. In a similar manner, this analysis was cross-sectional. Therefore, causal inference between BMD and phytate is limited. However, the major strengths of the study are the use of the simple and reproducible in-vitro model, the use of a DXA scan for measuring BMD, the control for dietary potential confounding variables, and that our results with post-menopausal women are completely in accordance with our results in vitro.

Finally, our results show that phytate, a natural product present legumes and cereals, inhibits HAP dissolution and that a high phytate consumption is associated with a higher BMD and t-score values of lumbar spines in postmenopausal women. This protective effect is likely due to phytate adsorption in HAP surfaces. Thus, we suggest that a diet rich in phytate may help to prevent or minimize bone-loss disorders, such as osteoporosis.

## 5. Conclusions

As a conclusion, we have demonstrated that phytate inhibits HAP dissolution in a concentration-dependent manner. The in-vitro effect of phytate on inhibiting HAP dissolution was similar to that of alendronate and greater than that of etidronate. Furthermore, the cross-sectional study indicated that a high phytate intake is associated with a normal-BMD (t-scores > −1). Our data suggest that phytate may have a role in osteoporosis prevention by adsorbing on the HAP surfaces. From this study follows the importance of the consumption of phytate-rich foods (nuts and legumes) to protect against the risk of osteoporosis. Nevertheless, further clinical trials and in-vitro studies are needed to investigate this potential role of phytate in bone-related diseases and its mechanism of action.

## 6. Patents

The patent entitled “Use of phytate as agent inhibiting dissolution of crystals of calcium salts for the prevention of osteoporosis (WO2007138147A1)” resulted from the work reported in this manuscript.

## Figures and Tables

**Figure 1 nutrients-13-02859-f001:**
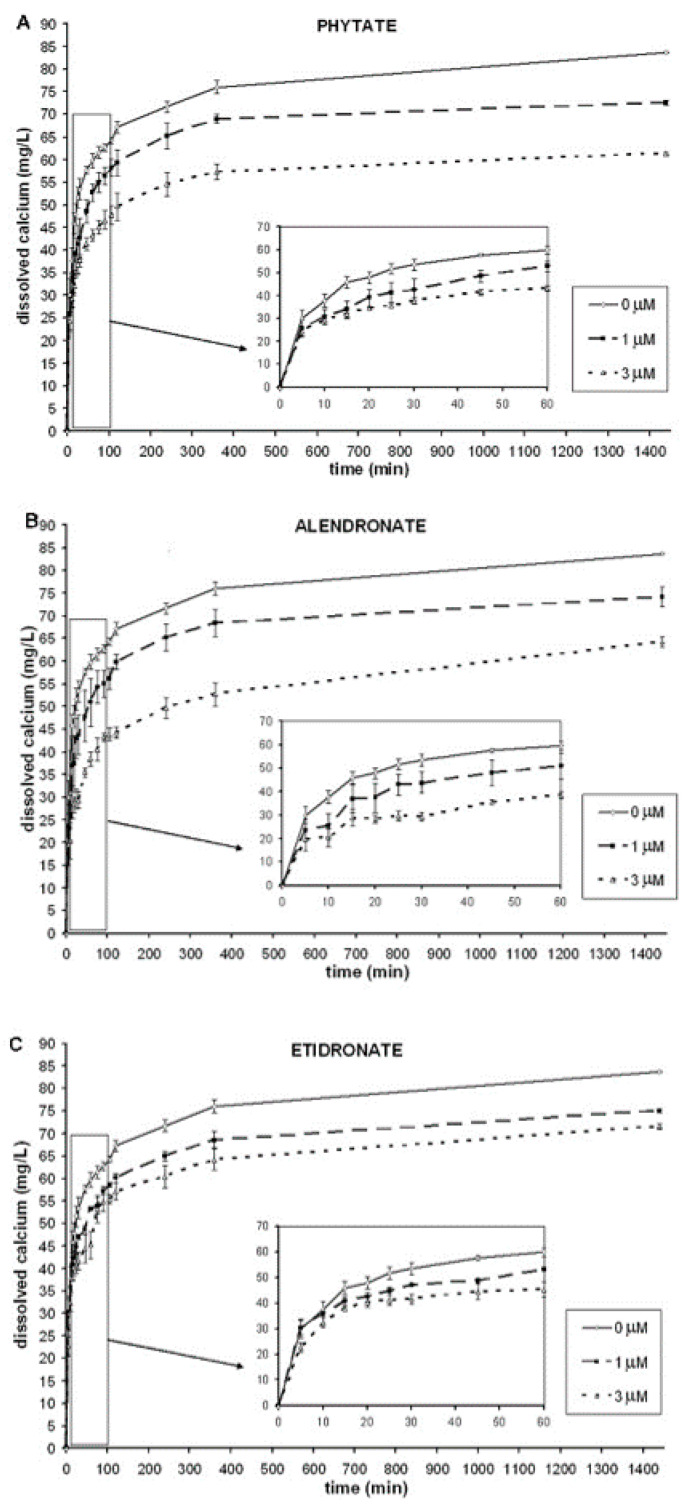
24 h-kinetic curves of the HAP dissolution using phytate, alendronate, and etidronate as inhibitors was incubated with nothing (control), phytate, alendronate, or etidronate (1 or 3 µM) to ensure that polyphosphates bound to HAP crystal surfaces. After that, the HAP-adsorbed drug was exposed to acid-driven dissolution. Aliquots were withdrawn at different times during 24 h and analyzed for the dissolved calcium: (**A**) phytate; (**B**) alendronate; (**C**) etidronate. All experiments were performed in triplicate. HAP: hydroxyapatite.

**Figure 2 nutrients-13-02859-f002:**
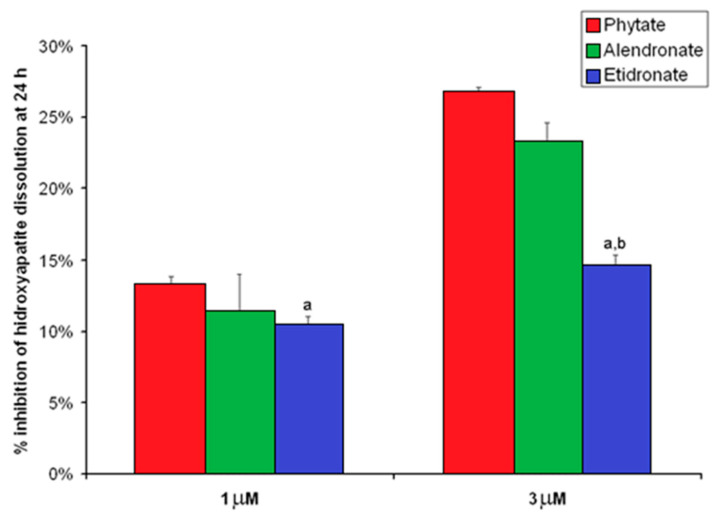
Percentage of inhibition of HAP dissolution by phytate, alendronate, and etidronate at 1 and 3 µM. Percentage of inhibition was calculated as it is indicated in the following formula “Percentage of inhibition = [mg calcium at 24 h for 0 µM–mg calcium at 24 h for 1 or 3 µM]/[mg calcium at 24 h for 0 µM] × 100”. Values are expressed as mean ± SE. Comparison were determined by ANOVA and Bonferroni as a post-hoc test. ^a^ *p* < 0.05 versus the value for the corresponding phytate concentration; ^b^ *p* < 0.05 versus the value for the corresponding alendronate concentration.

**Figure 3 nutrients-13-02859-f003:**
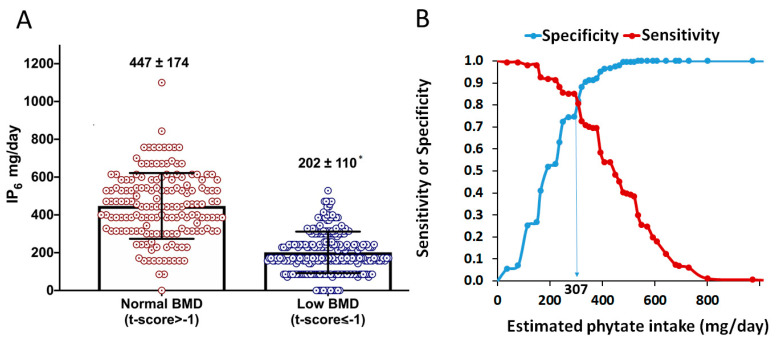
Phytate intake and BMD. (**A**) Estimated phytate intake for normal- and low-BMD groups. Values are expressed as mean ± SD. * *p*-value < 0.05 versus normal BMD calculated by *t*-test for independent samples. (**B**) Sensitivity (Se) and specificity (Sp) of estimated phytate intake (test variable) respecting to low-BMD (versus normal-BMD; state variable). The vertical line indicates the value that concurrently optimizes sensitivity (Se) and specificity (Sp), which corresponds to an estimated phytate intake lower than 307 mg/day (Se = 81.9%. Sp = 80.7%. Youden Index = 0.626).

**Table 1 nutrients-13-02859-t001:** Characteristics and bone parameters according to tertiles of estimated phytate intake (mg/day).

	Low IP6 Intake	Moderate IP6 Intake	High IP6 Intake	*p*-Value
	(0–199) mg/Day	(200–325) mg/Day	>325 mg/Day	
	(*n* = 145)	(*n* = 127)	(*n* = 147)	
*Anthropometric parameters*				
Age (years)	56.6 ± 3.1	55.4 ± 2.7 ^a^	55.0 ± 2.4 ^a^	<0.001
BMI (kg/cm^2^)	25.5 ± 4.2	25.5 ± 4.9	24.2 ± 3.7 ^a,b^	0.011
Hip circumference (cm)	106 ± 12	106 ± 13	102 ± 13 ^a^	0.019
Waist Circumference (cm)	89 ± 15	89 ± 15	86 ± 15	0.144
Body Fat Mass (%)	34.6 ± 9.1	34.2 ± 9.5	31.7 ± 9.5 **^a^**	0.018
Systolic Blood Pressure (mmHg)	118 ± 15	120 ± 15	119 ± 15	0.468
Diastolic Blood Pressure (mmHg)	75 ± 10	75 ± 10	73 ± 11	0.140
*Dietary parameters*				
Estimated Phytate Intake (mg/day)	117 ± 54	268 ± 37 ^a^	499 ± 129 ^a,b^	0.032
MedDiet Score (14-points)	8.2 ± 1.6	8.9 ± 1.5 ^a^	9.7 ± 1.9 ^a,b^	<0.001
High Adherence to Med Diet (≥9 points)	64 (44.1%)	81 (65.9%) ^a^	108 (73.5%) ^a^	<0.001
*Lumbar Spine L1–L4*				
Bone Mineral Density (g/cm^2^)	0.69 ± 0.07	0.75 ± 0.07 ^a^	0.87 ± 0.09 ^a,b^	<0.001
t-score	−1.70 ± 0.44	−1.31 ± 0.50 ^a^	−0.44 ± 0.72 ^a,b^	<0.001
Low-BMD (t-score ≤ −1)	132 (91.0%)	92 (74.8%) ^a^	30 (20.4%) ^a,b^	<0.001

Values are expressed as mean ± SD or frequency (percentage). *p*-value was calculated with ANOVA test and Bonferroni as a post-hoc test (quantitative data); or chi-square test (qualitative data). Statistics: ^a^ *p* < 0.05 versus corresponding value of low IP6 intake group; ^b^ corresponding value of moderate IP6 intake group. IP6:Myo-inositol hexaphosphate (phytate; IP6).

**Table 2 nutrients-13-02859-t002:** Comparison of anthropometric parameters and dietary characteristics between low-BMD (t-score ≤ −1) and normal-BMD (t-score > −1).

	Normal-BMD	Low-BMD	*p*-Value
	(t-Score > −1)	(t-Score ≤ −1)	
	(*n* = 161)	(*n* = 254)	
*Anthropometric parameters*			
Age (years)	55.0 ± 2.3	56.1 ± 3.1	<0.001
BMI (kg/cm^2^)	24.5 ± 4.3	25.3 ± 4.2	0.052
Hip Circumference (cm)	103 ± 13	106 ± 13	0.035
Waist Circumference (cm)	87 ± 15	89 ± 15	0.087
Body Fat Mass (%)	32.1 ± 9.7	34.3 ± 9.2	0.022
Systolic Blood Pressure (mmHg)	119 ± 15	119 ± 14	0.991
Diastolic Blood Pressure (mmHg)	73 ± 10	75 ± 10	0.027
BMD (g/cm^2^)	0.88 ± 0.07	0.70 ± 0.06	<0.001
t-score	−0.33 ± 0.54	−1.65 ± 0.38	<0.001
*Dietary food intake*			
Vegetables (g/day)	411 ± 217	351 ± 216	0.006
Fruits (g/day)	505 ± 258	417 ± 224	<0.001
Nuts (g/day)	114 ± 61	35 ± 35	<0.001
Legumes (g/day)	379 ± 193	312 ± 168	<0.001
Red Meat (g/day)	74 ± 76	76 ± 73	0.828
White meat (g/day)	110 ± 40	110 ± 41	0.123
Fish (g/day)	296 ± 153	271 ± 142	0.092
Olive Oil (g/day)	30 ± 13	30 ± 15	0.656
Butter (g/day)	1.7 ± 5.8	0.6 ± 2.3	0.020
Pastries and Sweets (g/day)	76 ± 89	66 ± 90	0.259
Carbonated Drinks (g/day)	81 ± 154	72 ± 135	0.522
Wine (g/day)	87 ± 170	68 ± 153	0.234
MedDiet Score. 14 points	9.5 ± 1.8	8.6 ± 1.7	<0.001

Values are expressed as mean ± SD or frequency (percentage). *p*-value was calculated with independent *t*-test (quantitative data) or chi-square test (qualitative data). BMD: bone mineral density.

**Table 3 nutrients-13-02859-t003:** Binary logistic regression of estimated phytate, legumes, and nuts intake with respect to low BMD (t-score ≤ −1, dependent variable) using normal BMD (t-score > −1) as reference (O.R. = 1).

	Crude O.R.	95% CI for (Crude O.R.)	*p*-Value	Adjusted * O.R.	95% CI for (Adjusted * O.R.)	*p*-Value
Estimated phytate intake						
Tertile 1 (0–200 mg/day)	1	(reference)	<0.001	1	(reference)	<0.001
Tertile 2 (200–325 mg/day)	0.29	(0.15–0.59)	0.001	0.32	(0.16–0.66)	0.002
Tertile 3 (>325 mg/day)	0.03	(0.01–0.05)	<0.001	0.03	0.01–0.06	<0.001
Legumes (per servings/week)	0.41	(0.31–0.52)	<0.001	0.44	0.33–0.59	<0.001
Nuts (per servings/week)	0.38	(0.31–0.46)	<0.001	0.39	0.32–0.47	<0.001

* Adjusted by age (years), BMI (kg/cm^2^), hip circumference (cm), waist circumference (cm), body fat (%), systolic blood pressure (mmHg), diastolic blood pressure (mmHg), MedDiet Score (14 points), vegetables (servings/week), fruits (servings/week), fish (servings/week), and butter (servings/week). Values are expressed as Odds Ratio (O.R.) and 95% Confidence Interval (CI).

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
