# Peer review of "Understanding the Protective Effect of Phytate in Bone Decalcification Related-Diseases"

_nutrients, 2021, doi:10.3390/nu13082859_

Round 1
Reviewer 1 Report
The manuscript “Understanding the protective effect of phyate in bone decalcification related-diseases “ is very interesting and brings important knowledge about phytate. It is well organized and describes original and potentially important results.
Author Response
We thank to the reviewer for his/her appreciation and constructive comments.

Reviewer 2 Report
Dear Authors,
Thank you very much for the opportunity to review this paper. In the light of the increasing women life expectancy, the importance of food choices (which may be important for health) seems to be
increasingly important. This issue is very important for the quality of life and population health. The job requires minor revision.
The comments are listed below:
The manuscript is well thought out and well written.
My great doubt concerns the lack of consideration in the analyzes (and in the introduction and discussion) of the importance of estrogen hormone levels and menarche age, which are highly important for
BMD/BMC. The beneficial effect of earlier menarche on bone tissue is well described a long time ago (https://repozytorium.amu.edu.pl/bitstream/10593/3454/1/04szkl.pdf). In the absence of estrogen concentration determinations, questionnaire data can be used. Also, BMI is another very important determinant of the bone mineralization state. If these variables cannot be included in the analysis, it is worth mentioning them in theoretical sections and study limitations.
l. 115+ In section 2.2.1. Subjects and Methods lack of a detailed description of the study group - number of respondents, average age, BMI. How many patients were excluded from the study? Do the authors
have other data on the lifestyle (e.g. smoking), marital status, fertility, number of pregnancies, e.t.c. of the surveyed women? If so, complete the table with the characteristics of the respondents.
l.146 - Specify whether the assumptions of the analysis of variance were met (which test) - complete in section 2.3. Statistical Analysis.
l. 138+ Include in the publication the measurement error of the technician / laboratory error.
l. 229 - In table 3 - Instead of Tertile 1 (0-200 mg / day) you should read Tertile 1 (0-199 mg / day). Parentheses should be closer to the numbers, e.g. (0.15 - 0.59) (NOT ( 0.15 - 0.59 )).
l. 366+ Minor technical errors in the literature - should be corrected. Please take a close look at the whole text and fill in the missing punctuation marks, spaces, etc.
Thank you for the opportunity to review this article.
Author Response
We thank to the reviewer for his/her appreciation and constructive comments, which have led to a much-improved version of our manuscript. All his/her comments/suggestions are addressed as detailed in this point-by-point reply and the changes in the manuscript have been highlighted.
